# Advancing Semantic Edge Detection through Cross-Modal Knowledge Learning

## ABSTRACT

Semantic edge detection (SED) is pivotal for the precise demarcation of object boundaries, yet it faces ongoing challenges due to the prevalence of low-quality labels in current methods. In this paper, we present a novel solution to bolster SED through the encoding of both language and image data. Distinct from antecedent language-driven techniques, which predominantly utilize static elements such as dataset labels, our method taps into the dynamic language content that details the objects in each image and their interrelations. By encoding this varied input, we generate integrated features that utilize semantic insights to refine the high-level image features and the ultimate mask representations. This advancement markedly betters the quality of these features and elevates SED performance. Experimental evaluation on benchmark datasets, including SBD and Cityscape, showcases the efficacy of our method, achieving leading ODS F-scores of 79.0 and 76.0, respectively. Our approach signifies a notable advancement in SED technology by seamlessly integrating multimodal textual information, embracing both static and dynamic aspects.

## CCS CONCEPTS

• **Computing methodologies** → *Computer vision tasks*; **Shape inference**.

## KEYWORDS

Semantic Edge Detection, Contour Detection, Low-level Vision, Deep Convolutional Neural Networks

## 1 INTRODUCTION

Semantic edge detection (SED) [31] aims to identify and classify object boundary pixels simultaneously, necessitating high classification accuracy and strategies to handle the skewed edge-to-background pixel ratio. This task is crucial in multimedia applications, including image and video understanding, content-based retrieval, and augmented reality. Deep learning techniques have been extensively utilized in SED research [1, 21, 31, 32] within the multimedia domain, employing varied encoder-decoder architectures with sophisticated backbones, post-processing methods, and hybrid models to address inherent challenges. However, the effectiveness of these methods is often hindered by the quality of

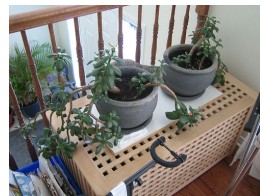

(a) An example from SBD[11]  (b) Interest Object Boundary

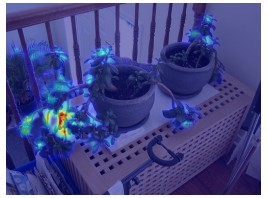
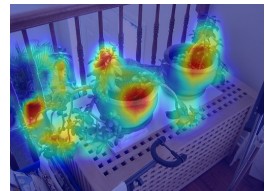

(c) Casenet Original Attention  (d) Improved via Our Approach

**Figure 1: Illustration of our method enhancing SED tasks. Figures demonstrate Casenet's [31] attention calibration, transitioning from single-modal to cross-modal learning, effectively covering regions of interest and aligning with target objects. We provide additional evidence of how our method improves feature representation, attention maps, and boosts model performance in experimental section.**

semantic edge labels [11]. The derivation of SED labels from semantic segmentation (SS) labels, which often lack precise boundary information for objects, leads to inaccuracies in SED labels. This deficiency results in missing boundary pixels, causing inaccurate regression effects in single-modality learning methods, highlighting the importance of advancements in this area to enhance multimedia applications further.

SED essentially involves classifying individual pixels, similar to semantic segmentation and medical image segmentation. To address the challenge, we're exploring recent advancements in related domains. For example, CLIP-UM [20] revolutionizes medical segmentation by integrating the CLIP text encoder [24] with an encoder-decoder framework, resulting in a significant boost in model accuracy, which ranked first in Mediacal Segmentation Decathlon Competation. Meanwhile, LSeg [17] pioneers zero-shot learning using CLIP's methodology, highlighting the transformative impact of text data integration on encoder-decoder models. SAM (Segment Anything) [14] is a powerful AI model that segments any object in an image using various input prompts such as text, points, and masks to ensure accurate results with ease.

While language-driven models such as CLIP-UM and LSeg have excelled in semantic segmentation, even ranking first in competitions like [20], their direct application to SED encounters unique hurdles. One significant challenge lies in handling labels, which are more dispersed in SED. Compared to SS labels in datasets like

PASCAL VOC, SED deals with significantly fewer foreground pixels in the dataset of SBD (30.5% VS. 3.2%). This underscores the importance of precise pixel classification and accurate edge detection amidst predominantly background areas. Applying techniques designed for SS to SED exacerbates the issue, highlighting the need for novel and tailored methods to address SED's specific challenges.

To address these challenges, we propose a novel cross-modal contour detection method for SED. Inspired by CLIP-UM and LSeg, our approach learns from both images and text, facilitating comprehensive contour representations. However, we've tailored our model with unique designs to better suit SED's distinctive traits: Firstly, we explore dynamic language information, detailing objects within each image and their relationships, departing from prior language-driven methods. Secondly, we integrate a fusion technique to enhance cross-modality information integration. By encoding this diverse information, we create integrated features, leveraging semantics to refine high-level image features and mask encodings. Our method significantly improves SED accuracy compared to previous models, focusing on three key contributions:

— We propose leveraging text information to enhance semantic edge detection through a novel cross-modality framework. This approach effectively guides the generation of high-level image features and final mask encodings, significantly enhancing feature representation and model performance.

— We propose encoding multiple language information to demonstrate the robustness of the proposed framework. We find that increasing the textual content, such as object names and descriptions of their relationships in each image, also enhances model performance.

— Through extensive experiments on SBD and CityScape datasets, we demonstrate our method's significant outperformance of previous state-of-the-art (SOTA) detectors. Moreover, we investigate and discuss the potential reasons text features enhance edge detector performance and showcase our framework's zero-shot ability to identify edge categories.

## 2 RELATED WORK

In this section, we provide a concise overview of recent advancements in semantic edge detection.

Traditional edge detection has flourished in the field of digital image processing, primarily by designing various image gradients[3, 10, 15, 23]. In recent years, SOTA edge detectors have predominantly utilized deep convolutional neural networks (CNNs) [5–8, 16, 26, 30, 33], achieving comparable performance with humans. However, due to limited annotated labels, there is still ample room for improvement in the task.

Challenges in semantic edge detection, as showcased in Casenet [31], mainly revolve around the complex task of accurate boundary detection and class assignment for edge pixels. Compared to traditional edge detection[5–8, 16, 26, 30, 33], this task requires additional classification of pixel categories. Current edge annotations often originate from semantic segmentation datasets like Pascal VOC [9], which do not specialize in edge detection, and thus create potential inaccuracies during interpretation. Seal [32] tackled this by optimizing the process to diminish errors stemming from inadequate annotations. In a similar vein, Steal [1] presented the concept

of end-to-end edge perfection through diligent post-processing, while DDS [21] enhanced edge localization via quadruple auxiliary features, addressing the issue of unreliable edge annotations.

The integration of linguistic elements into computational models has recently gained remarkable attention. The CLIP model [24], with its contrastive training of dual transformer-based encoders on a rich collection of text-image pairs, stands at the forefront of this field. This model's text encoder is particularly adept at extracting versatile features. Capitalizing on these capabilities, novel dense prediction methodologies such as LSeg [17], SegCLIP [22], and CLIP-UM [20] employ the CLIP framework to integrate textual data with image characteristics, registering impressive outcomes in dense prediction tasks. SAM [14] encodes multiple modalities of information, including language, bounding boxes, masks, and more. It demonstrates remarkable ability to detect and segment new objects not present in training datasets.

## 3 LANGUAGE-DRIVEN EDGE DETECTION

SED labels often lack accuracy due to missing boundary pixels from semantic segmentation labels, leading to inaccurate regression effects in single-modality learning methods. Meanwhile, language-driven methods offer advantages in accuracy and zero-shot detection, addressing the limited research in SED. Therefore, in this study, we propose leveraging language-driven approaches to enhance SED. We detail our proposed method in the following section.

### 3.1 Overall Framework

Our model leverages an encoder-decoder structure that merges multi-resources features and generates pixel-precise masks, meeting the demands of semantic edge detection (SED). Diverging from standard edge detectors, it integrates dual encoders for images and language, along with two novel fusion techniques to enhance cross-modality information. The overall structure is shown in Fig. 2.

In greater detail, our proposed method amalgamates the visual and textual modalities to boost the semantic edge detection capability. The model accepts an image $I \in \mathbb{R}^{H \times W \times 3}$ and pairs it with relevant textual descriptions $T$. It generates a mask $P \in \mathbb{R}^{H \times W \times N}$ as the output. The edge detection mechanism consists of two parts: an encoder $\mathbf{E}$ and a decoder $\mathbf{D}$. Within our framework specifically, the encoder is bifurcated into an image encoder $\mathbf{E}_I$ and a text encoder $\mathbf{E}_T$, both subsumed under the comprehensive framework equation. The framework can be expressed as:

$$P = \mathbf{D}(\mathbf{F}(\mathbf{E}_I(I), \mathbf{E}_T(T))). \tag{1}$$

To efficiently fuse image and text features, we propose a two-step fusion process: first, combining the two features at the tops of both encoders, and then feeding the fused feature to the decoder. Here, $\mathbf{F}$ represents the proposed cross-modality fusion module to complete the above steps. The subsequent content provides detailed descriptions of encoders and our decoder.

*3.1.1 Image encoder.* In our research, the encoder-decoder network plays a crucial role in generating multi-scale and multi-level side features. Specifically, we leverage the highest-level feature for cross-modality feature fusion. To demonstrate the generalization performance of our proposed method, we utilize various encoders

**Figure 2: Overview of the proposed edge detector: We propose leveraging cross-modality information to refine both high-level image features and final mask encodings, aiming to enhance model performance. Notably, we omit the classification head for optimization purposes.**

including VGG-16 [25], EfficientNet-b7 [27]. The highest performance achieved in our experiments is with the EfficientNet-b7 backbone, which we utilize for SOTA comparisons.

*3.1.2 Text encoder.* The text encoder $E_T$ converts label texts into numerical features, associating N-label words with N continuous vectors. For instance, in the SBD dataset with 20 label classes, we input these 20 label texts into a text encoder, yielding the corresponding text encodings. Similar to the image encoder, various text encoders can be employed for this purpose. In this study, we employ the CLIP text encoder. It's important to note that the text encoder remains frozen throughout all experiments, indicating that it is not updated during the training process.

*3.1.3 Decoder.* The decoder **D** combines mask encoding with backbone side features in the refinement path. Our strategy uses separate convolutional layers to smooth side features and mask encoding pairs. Results are combined, and a third convolutional layer reduces channel count. A deconvolutional layer is then used for resolution upsampling. The decoder repeats this process until the resolution of the feature matches the input image. Finally, a convolutional layer adjusts output features to align with the dataset's label count.

*3.1.4 Fusion Modules.* Our key focus is the fusion module **F**, which is synonymous with our knowledge learning module. This prompts us to provide a detailed explanation of this crucial element. Following that, we'll introduce the various learning resources before diving into the module's specifics.

## 3.2 Cross-modality Knowledge Learning

As mentioned earlier, we explored feature fusion across text and image features. However, we did not delve into the composition of these data, such as the required format of text data for a text encoder. We address this question at the beginning of this subsection.

**Static Language Information** For text input, language-driven models typically utilize the CLIP text encoder, which leverages contrastive language-image learning and yields high-quality language features for learning. We also employ this approach and follow the method to structure our data. The procedure involves using a prompt template to concatenate a set of text labels. Taking the SBD dataset as an example, it comprises 20 categories, each associated with a corresponding text label such as *['dog', 'cat', ..., 'car']*. We use the template *'a photo of a '* to generate the input text set, resulting in *['a photo of a dog', 'a photo of a cat', ..., 'a photo of a car']*. Subsequently, we feed this text set into the CLIP text encoder to obtain the text embeddings. It is noteworthy that these text embeddings remain unchanged for each input image, which we would call it as Static Language Information.

**Dynamic Language Information** We are aware that during model training, the input image varies among each other with each iteration. However, for most language-driven models like LSeg and CLIP-UM, the input language information remains unchanged (static), typically consisting of dataset label texts. This raises the question of *whether incorporating dynamic language information, such as object names that emerge in each image, can enhance contour representation and further improve model performance.*

This question is crucial because dynamic language information offers more semantic details than static language information, providing richer context for the image. In this study, we aim to explore the impact of dynamic language information on an encoder-decoder model. Beyond including object names in each image, we aim to encode their relationships, constructing a knowledge graph to represent objects and their interrelations. This knowledge graph consists of nodes (entities or objects) and edges (relationships among objects), which will be integrated into the edge detector along with dataset label texts for training and inference.

Thanks to the advancements in Large Vision Models (LVMs) [19, 34], as well as open-source Large Language Models [13, 28], implementing our idea to extract knowledge graphs from images has become more feasible. We proceed in two steps: First, we utilize an LVM, such as the MiniGPT4 [34], to describe the Fig.1a by prompting it with:

*"Please describe the image."*

The LVM generates a textual description based on the image content, such as: *"The image shows a wooden rack with two potted plants on it. The plants are small and have green leaves. The rack is made of wood and has a rusted metal grate on the bottom. There is a wooden staircase leading up to the rack. The room is dimly lit, and there are no other objects in the image."* Subsequently, we input this text into a Large Language Model (LLM) such as Mixtral 7b, using the following prompt:

*"Extract the entities and relations from the sentence, ignoring adjectives and adverbs, and output the result strictly in the format of '(entity1, entity2, their relation)': item['output']"*

Here, entity1 and entity2 represent objects, while their relation indicates the edge between nodes. Applying this procedure to the provided text, we obtain the final output along with the corresponding knowledge graphs: *([wooden rack], [two potted plants], [has, contains])(wooden rack, [rusted metal grate], [has, part])(room, [image], [is, part of])(image, [wooden staircase], [has, part])(wooden rack, [wood], [is made of, composed of])(wooden rack, [dimly lit room], [is located in, in]).* Before training, we store all these knowledge graphs as text for each image. During training, we pair each knowledge graph with the dataset label text and the input image.

Please note that the extracted knowledge graphs may contain noise, primarily due to the limitations of the LLM's capacity, which performs entity recognition and generates specific forms. Due to resource constraints, we utilized an NVIDIA 3090 card with 24GB memory, which can only accommodate the Mixtral 7B model [13]. As a result, the generated dynamic language information may not meet our expectations in terms of quality. Nevertheless, this information remains valuable for SED purposes.

**Image Information** The input image format remains the same as that of traditional edge detectors [8, 30]. We feed a full-size RGB image and expect a multi-channel output of the same size.

*3.2.1 Knowledge Learning and Feature Fusion.* When conducting cross-modal feature Learning, our goals are twofold: firstly, to enhance the quality of generated image features by integrating new features from cross-modal fusion; secondly, to ensure that the fused features directly improve the mask quality without losing information during transmission. To achieve this, we propose two fusion points on the encoder-decoder model: one at the highest-level feature generation stage to enrich feature representation, and another at the final image feature location to enhance mask encoding quality, preventing information loss from prolonged transmission.

**High-level Image Feature Enhancement** We begin by examining the initial fusion point, where we pinpoint the output feature location of an encoder's block-5. This spot is pivotal as it corresponds to the generation of the highest-level image features, which are subsequently converted into mask encodings in the decoder.

Enhancing the quality of this feature would consequently bolster the model's performance.

Assuming $T_S$, $T_D$ and $I$ represents a dataset label text (static language information), a knowledge-graph text (dynamic language information) and an image, We first encode these inputs via text encoders such as CLIP and image encoders such as VGG16, to get the text features $f_S$, $f_D$ uniformly referred to as $f_T$ and the image feature $f_I$. we first define a novel cross-modality feature fusion method, which we call it attention-calibration fusion, since it mainly relies on first fusing text and image features, then using the calibrated features to refine the high-level image feature.

The attention-calibration fusion is following: Feature shape alignment, feature fusion and feature recalibrate. Addressing differences in feature shapes $f_T$, $f_I$ from independent encoders, the consolidation process involves compressing features into channel descriptors $f_{T-aligned}$, $f_{I-aligned}$ with global average pooling, synchronizing them through channel adjustment, and activating the aligned features with ReLU. We express this process of function as Eq. 2

$$
\begin{aligned}
f_{T-\text{aligned}} &= \text{AvgPooling}_{1\times1}(f_T) \\
f_{I-\text{aligned}} &= \text{AvgPooling}_{1\times1}(f_I) \\
f_{I-\text{aligned}} &= \text{ReLU}(\text{FC}(f_{I-\text{aligned}})) \\
f_{\text{fusion}} &= f_{T-\text{aligned}} \odot f_{I-\text{aligned}} + f_{I-\text{aligned}} \\
f_{\text{weights}} &= \text{Sigmoid}(\text{FC}(f_{\text{fusion}})) \\
f_{I\text{-calibration}} &= f_{\text{weights}} \odot f_I
\end{aligned}
\tag{2}
$$

The aligned feature descriptors then enter a fusion process where the feature fusion function intricately blends text and image features through an operation that couples element-wise multiplication with a residual addition. Post-fusion, the calibrated high-level image feature $f_{I\text{-calibration}}$ is determined by a gating mechanism, which scales the feature values with a sigmoid function and a fully connected layer to modulate the discriminative significance of the original high-level image feature, enhancing critical semantics while diminishing non-essential information.

At the location of an encoder's block-5, we utilize the fusion function 2 twice for both static and dynamic language features $f_S$ and $f_D$. We separately generate two fusion features. Then, we use a convolution layer to directly fuse these two fusion features as follows:

$$
\begin{aligned}
f_{I\text{-CM}} = &ReLU(\text{Conv-BN}(\text{Attn-Calib}(f_S, f_I)) \\
&+ \text{Conv-BN}(\text{Attn-Calib}(f_D, f_I)))
\end{aligned}
\tag{3}
$$

Here, Attn-Calib represents the fusion function 2, and Conv-BN denotes a convolutional layer followed by a BatchNorm layer. The generated new $f_{I\text{-CM}}$ (CM denotes Cross Modality) replaces its original counterpart $f_I$ and will be fed into the decoder **D**. The whole process is show in the central module of Fig. 2.

**Final Mask-encoding Refinement** The other fusion location we select is the final mask-encoding. To justify this choice, we need to ensure that the fused features directly enhance mask quality without losing information during transmission, as this location is distant from the encoder's block-5. Unlike the high-level features of block-5, final mask encodings exhibit significant differences: they have the same size as the input image, whereas the output of block-5 has the smallest size. Moreover, final mask encodings contain

both high-level information and object low-level details, whereas the output of block-5 only contains high-level information. These distinct characteristics impose new requirements for effectively fusing language information and mask encodings.

We plan to leverage cross-modality features to refine the final mask-encodings. Our method is straightforward: we apply a residual connection from the feature $f_{I\text{-CM}}$ and directly fuse the final mask-encoding using the Attn-Calib function Eq. 2:

$$f_{I\text{-final}} = \text{Attn-Calib}(f_{I\text{-CM}}, f_{I\text{-final}}) \qquad (4)$$

Through the residual connection, we ensure that cross-modality knowledge can be enhanced in the final mask-encoding, further improving model performance.

Note that our cross-modal framework doesn't rely solely on either static or dynamic language information. We can use just one type of text with image data and still achieve better performance compared to using images alone. In this scenario, we employ the Attn-Calib process Eq. 2 to fuse the image modality with either $f_S$ or $f_D$, obtaining $f_{I\text{-calibration}}$. This feature guides the refinement of the final mask-encoding directly using Eq. 4. We demonstrate multiple experiments to showcase our framework's flexibility in the next section. The whole process is show in the right module of Fig. 2.

After obtaining the refined final mask-encoding, we use a $3 \times 3$ Conv layer as a classification head to generate the final mask, which serves as our model's output. Subsequently, we utilize the CrossEntropy loss to train the entire model. In the following section, we will delve into the specifics of our experiments.

## 4 EXPERIMENTS

### 4.1 Datasets and Implementation Details

**Datasets** We use two datasets to evaluate semantic edge detection: SBD [11] and Cityscapes [4]. SBD has 11,355 images from PASCAL VOC2011, with 8,498 for training and 2,857 for testing (20 categories). Cityscapes provides pixel-level data for 5,000 street scene images; we used 2,975 for training and 500 from the validation set as the test set (19 categories).

**Implementation Details** Training employed the Adam optimizer with the following settings: epochs: 200, learning rate: 1e-4, weight decay: 5e-4. For SBD, images smaller than $480 \times 480$ were padded and augmented by adjusting the overall image scale (0.5 to 1.5) with random cropping. For Cityscapes, images of size $2048 \times 1024$ were cropped into $1024 \times 512$ parts, with no scale adjustments made. Both datasets underwent data augmentation, including random flips. It's worth noting that for state-of-the-art (SOTA) comparisons, we used EfficientNet-b7 as the backbone to demonstrate the best performance. For the ablation study, VGG16 was chosen as the backbone to accelerate the training process.

**Evaluation Metric** We adopted the Optimal Dataset Size (ODS) as the evaluation metric, following prior works [2, 12, 21, 31, 32]. The ODS scores for individual categories and the average ODS (mean ODS) across all categories will be presented.

### 4.2 State-of-the-Art Comparisons

**Results of SBD.** We conducted experiments on the SBD dataset, comparing CASENet [31], STEAL [2], DDS [21], and our method.

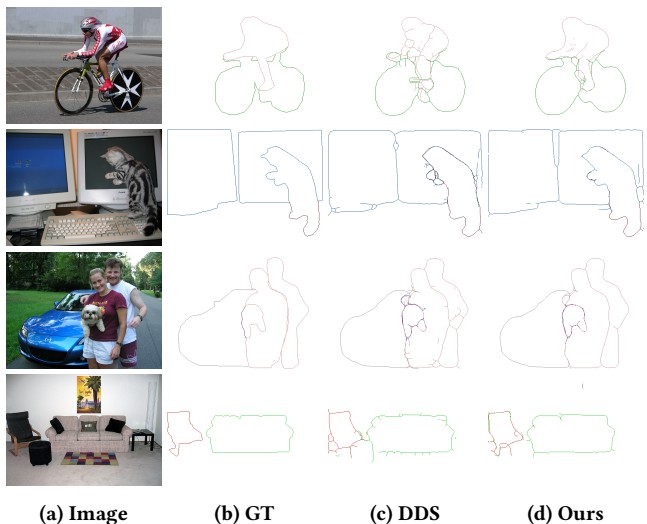

| (a) Image | (b) GT | (c) DDS | (d) Ours |

**Figure 3: SOTA comparisons on the SBD dataset show that our method preserves fine object details while exhibiting fewer artifacts, demonstrating its effectiveness.**

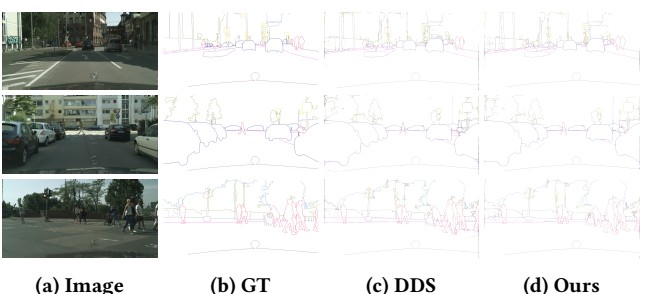

| (a) Image | (b) GT | (c) DDS | (d) Ours |

**Figure 4: SOTA comparisons on the CityScapes dataset consistently demonstrate that our method excels in preserving fine details.**

CASENet, as the pioneer, laid the groundwork for subsequent advancements. STEAL, DDS, and related methods evolved from the CASENet framework. All the methods were initially pretrained on the COCO dataset [18] before training on the SBD. Since we employ EfficientNet-b7 to get the best performance, we also trained a EfficientNet-b7 version of DDS for fair comparison. For quantitative results, refer to Table 3, and for qualitative results, refer to Figure 3.

In terms of quantitative results in Table 3, our approach achieves state-of-the-art performance with a mean ODS of 79.0, significantly surpassing DDS (77.0 using EfficientNet-b7), STEAL (75.8), and others. We present two versions of the proposed method: Ours-Static-EFF, trained on EfficientNet-b7 using only dataset label text, and Ours-Both-EFF, using both dataset label text and the knowledge graph. We observe that with the addition of language information, our performance surpasses methods tailored for better regression of a single image modality.

Regarding qualitative results in Fig. 3, key areas where our method delineates itself from DDS include the cyclist, where our

| VGG-16 | Fusion Module 1 | Fusion Module 2 | Mean-ODS |
|:------:|:---------------:|:---------------:|:--------:|
| ✓ | | | 64.9 |
| ✓ | | ✓ | 70.6 |
| ✓ | ✓ | ✓ | 72.5 |

**Table 1: Effectiveness of Model Components**

| VGG-16 | Static Language | Dynamic Language | Mean-ODS |
|:------:|:---------------:|:----------------:|:--------:|
| ✓ | | | 64.9 |
| ✓ | ✓ | | 70.6 |
| ✓ | | ✓ | 71.2 |
| ✓ | ✓ | ✓ | 72.5 |

**Table 2: Effectiveness of Language Information**

method accurately captures the overall contour including minute details like sleeves and pedals which seem blurry in DDS. For the cat between the table and keyboard, our method captures the intricate details such as ears and limbs which appear foggy in DDS. While portraying the car, our method aligns better with GT, especially while capturing the lower half such as wheels and car-bottom shadows where our method outperforms DDS. Finally in the furniture scene, our method excels DDS in outlining the edges and shapes of furniture, especially chairs and tablet, offering richer detail. Overall, our method exhibits superior performance in capturing object shapes and detailed edges, and in handling complex contextual information in the environment compared to DDS.

**Results of Cityscapes.** In the case of Cityscapes, we compare our method with CASENet and DDS. Both methods are pretrained on COCO. The results can be seen in Table 4 and Figure 4.

In Table 4, the scores across various object categories indicate the accuracy of each method in correctly identifying and locating corresponding objects' boundaries in the images. Our method exhibits superiority across most categories, with the highest average score of 76.0, demonstrating its overall supremacy. However, in categories like train and motorcycle, where DDS exceeds, there are potential areas for improvement. In summary, this table underscores the efficacy of our method in the CityScapes dataset, as it clearly outperforms in multiple object categories.

From Fig. 4, our method outperforms DDS in accuracy and detail across various challenging scenarios. It excels at road delineation, accurately marking boundaries and preserving intricate building details. Additionally, our method demonstrates superior vehicle recognition capabilities, effectively handling variations in size and color. Unlike DDS, our method aptly detects road sign elements such as zebra crossings and signboards, ensuring comprehensive result completeness.

## 4.3 Ablation Study

In this study, all experiments are conducted with a VGG16-based U-net edge detector on the SBD dataset.

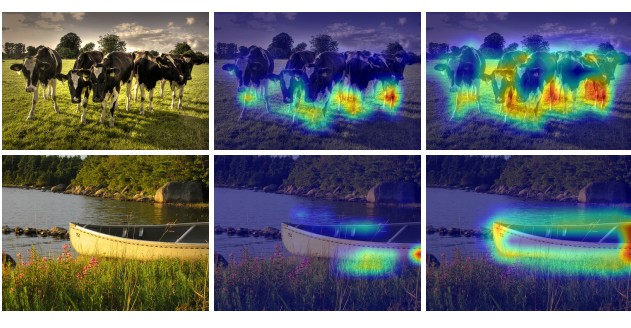

| (a) Images | (b) Original attention | (c) Using our method |
|:----------:|:----------------------:|:--------------------:|

**Figure 5: Our method improves models' attention maps, as seen in the examples of Ours (top) and DDS (bottom). Incorporating language information prompts the models to focus on interesting objects.**

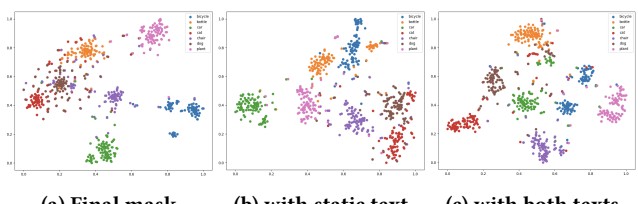

| (a) Final mask | (b) with static text | (c) with both texts |
|:--------------:|:--------------------:|:-------------------:|

**Figure 6: t-SNE visualization of final mask encodings with the addition of language features.**

*4.3.1 The effectiveness of model components.* From a model components perspective, our model is divided into three parts: a VGG16-based U-net edge detector, the first fusion module which applies Eq. 2 and Eq. 3 to generate fusion features, and the second fusion module which utilizes Eq. 4 to refine the final mask-encoding. We present the results in Table 1.

The baseline VGG16-based U-net edge detector achieved 64.9. Adding Fusion Module 2, which introduces static language information, significantly improves performance to 70.6. Incorporating Module 1, representing knowledge graph information, achieves 72.5. These findings highlight each component's effectiveness, but does static information outperform dynamic language information? Further analysis is conducted next.

*4.3.2 The effectiveness of language information.* In this experiment, we varied input language types to assess their impact. Results in Table 2 indicate that both dynamic and static texts enhance performance, with dynamic text slightly outperforming static. Combining both yields the best results, highlighting the effectiveness of utilizing multiple language inputs.

*4.3.3 The effectiveness of cross-modality information learning: improvements to model's attention, feature representation, and model performance.* We previously claimed that cross-modality learning improves attention and feature representation in models. Here, we conduct two experiments to demonstrate this. Firstly, we use two detectors, VGG16 U-net and DDS, to showcase the enhancement of high-level features. We extract features from their VGG16's block5

| Methods | aero | bike | bird | boat | bottle | bus | car | cat | chair | cow | table | dog | horse | mbike | person | plant | sheep | sofa | train | tv | mean |
|---|---|---|---|---|---|---|---|---|---|---|---|---|---|---|---|---|---|---|---|---|---|
| CASENet [31] | 83.3 | 76.0 | 80.7 | 63.4 | 69.2 | 81.3 | 74.9 | 83.2 | 54.3 | 74.8 | 46.4 | 80.3 | 80.2 | 76.6 | 80.8 | 53.3 | 77.2 | 50.1 | 75.9 | 66.8 | 71.4 |
| SEAL [32] | 84.9 | 78.6 | 84.6 | 66.2 | 71.3 | 83.0 | 76.5 | 87.2 | 57.6 | 77.5 | 53.0 | 83.5 | 82.2 | 78.3 | 85.1 | 58.7 | 78.9 | 53.1 | 77.7 | 69.7 | 74.4 |
| STEAL [2] | 85.8 | 80.0 | 85.6 | 68.4 | 71.6 | 85.7 | 78.1 | 87.5 | 59.1 | 78.5 | 53.7 | 84.8 | 83.4 | 79.5 | 85.3 | 60.2 | 79.6 | 53.7 | 80.3 | 71.4 | 75.6 |
| DDS [21] | 86.7 | 79.6 | 85.6 | 68.4 | 74.5 | 86.5 | 81.1 | 85.9 | 60.5 | 79.3 | 53.5 | 83.2 | 85.2 | 78.8 | 83.9 | 58.4 | 80.8 | 54.4 | 81.8 | 72.2 | 76.0 |
| DDS-EFF | 86.1 | 78.0 | 86.3 | 69.9 | 73.1 | 86.5 | 80.5 | 86.8 | 64.2 | 85.7 | 53.9 | 85.6 | 86.1 | 79.7 | 82.5 | 58.5 | 84.3 | 60.7 | 83.0 | 69.4 | 77.0 |
| Ours-Staic-EFF | 88.3 | 80.5 | 86.6 | 68.4 | 72.9 | 87.6 | 82.4 | 87.3 | 63.1 | 81.3 | 53.5 | 84.5 | 86.0 | 80.6 | 84.3 | 57.1 | 84.2 | 60.5 | 83.4 | 72.5 | 77.6 |
| Ours-Both-EFF | 87.9 | 80.2 | 87.1 | 70.0 | 75.9 | 86.6 | 83.1 | 88.1 | 65.2 | 86.4 | 56.7 | 86.8 | 88.6 | 82.0 | 84.3 | 62.7 | 87.8 | 59.9 | 85.1 | 74.6 | 79.0 |

Table 3: State-of-the-art comparisons on the SBD dataset. The proposed approach achieved the best performance. All the methods are pre-trained on the COCO dataset.

| Methods | road | s.walk | build. | wall | fence | pole | t-light | t-sign | veg | terrain | sky | person | rider | car | truck | bus | train | motor | bike | mean |
|---|---|---|---|---|---|---|---|---|---|---|---|---|---|---|---|---|---|---|---|---|
| CASENet | 86.6 | 78.8 | 85.1 | 51.5 | 58.9 | 70.1 | 70.8 | 74.6 | 83.5 | 62.9 | 79.4 | 81.5 | 71.3 | 86.9 | 50.4 | 69.5 | 52.0 | 61.3 | 80.2 | 71.3 |
| DDS | 89.7 | 79.4 | 80.4 | 52.1 | 53.0 | 82.4 | 81.9 | 80.9 | 83.9 | 62.0 | 89.4 | 86.0 | 77.8 | 92.3 | 59.8 | 74.8 | 55.3 | 64.4 | 77.4 | 74.9 |
| Ours | 90.3 | 80.2 | 80.9 | 51.3 | 55.5 | 82.6 | 81.7 | 82.6 | 83.9 | 62.6 | 89.1 | 86.1 | 76.2 | 91.8 | 59.0 | 79.8 | 70.1 | 63.4 | 76.7 | 76.0 |

Table 4: In the CityScapes dataset, our approach also attains the leading performance. All the methods are pre-trained on the COCO dataset.

and fuse them with language texts using our module, observing changes in attention. Images of a herd of cows and a boat by the lakeside are shown. The original attention column depicts the initial focus points of the algorithms, with highlighted areas indicating the focus. In the last column, shifts in highlighted areas indicate the model's altered focus via our method. This new focus, driven by linguistic information, targets specific interesting objects in the scenes.

For our second experiment, we analyze the feature representation of final mask encodings before and after refinement on the U-net's final mask-encodings. Using t-distributed stochastic neighbor embedding (t-SNE) [29], we visually represent the data structural changes of the final mask-encodings. The scatter plots show six categories from SBD after dimension reduction by t-SNE. The first plot displays the data structure of original U-net's final mask-encoding, indicating a lack of clear category recognition by the model. However, in the second plot, adding the dataset label text, distinct clusters emerge, particularly for 'plants' and 'bicycles', suggesting that adding static text improves category recognition. The final plot, adding with both dataset label text and knowledge graph text, shows a similar distribution to the original, but with some discernible clusters, indicating the model begins to grasp category features with both static and dynamic text. Though the desired outcome isn't fully achieved, a progressive trend suggests that including text enhances category recognition, improving classification performance.

## 4.4 Unveiling the Utility of Language Information in SED

After reading our work, one may raise a question: " why language information is useful for edge detection? what factor to finally influence the edge detector, so their performance is improved?". In this part, we set experiments to answer the question.

We first examine whether an edge detector is sensitive to the content of a text. Our aim is to ascertain whether an edge detector truly comprehends the text. To keep the experiment simple and clear, we keep the model structure, yet only utilize static language information. However, we input three different types of texts: the original dataset labels "a photo of a [label]" (Baseline A), the only text prompt "a photo of a" without any text labels (Baseline B), and a set of random strings (Baseline C). By comparing the latter two scenarios, we seek to understand the extent to which losing the content of the labels affects performance, and whether the prompt "a photo of a" significantly benefits model performance. The results are shown in Table .

Analyzing the results, we observe that all three types of text significantly contribute to enhancing model performance, even when using random strings. A notable trend is the progressive improvement in model performance with the augmentation of effective textual content. Notably, while the baseline U-net achieves a modest 64.9, the relatively weakest performer, Baseline C, achieves a commendable 70.1, indicating a substantial enhancement. We hypothesize that the incorporation of language features into the edge detector acts as a regularization mechanism, mitigating overfitting during training. This additional signal aids in optimizing the model's performance, where the model perform less sensitive to the content of the text. Moreover, the benefits of incorporating language features extend beyond regularization; later, we will demonstrate the model's newfound zero-shot capability upon their inclusion.

As we see performance improvements, a new question arises: Can any new text consistently boost performance by 5 points, regardless of fusion module design? Or, more succinctly, how much impact does fusion module design have? To investigate, we conducted three experiments: VGG16 U-net with single-modality regression (baseline 1), baseline 1 with a basic final-mask-encoding refinement module incorporating static language information (baseline 2). This module aligns Clip text encoder's features with U-net's final mask-encoding and uses a simple $3 \times 3$ Conv layer for fusion. Therefore, in this naive refinment, we only use a conv layer to perform refinement. The third baseline is the U-net with our mask-encoding refinement on static language information. The results are shown in Table 6.

| VGG-16 | Prompt Templates | | | Mean-ODS |
|---|---|---|---|---|
| | A | B | C | |
| ✓ | ✓ | | | 70.6 |
| ✓ | | ✓ | | 70.2 |
| ✓ | | | ✓ | 70.1 |

**Table 5: Influences of different prompts to model performance.**

| VGG-16 | Naive refinement | Our refinement | Mean-ODS |
|---|---|---|---|
| ✓ | | | 64.9 |
| ✓ | ✓ | | 65.3 |
| ✓ | | ✓ | 70.6 |

**Table 6: Importance of fusion module design**

Analyzing the results, we observe that while the naive refinement module yields a mere 0.4-point improvement with identical text input, the proposed mask-encoding refinement achieves a substantial 5.7-point enhancement. To clearly identify the influencing factors, we deliberately omitted the high-level feature improvement module. These findings underscore the critical importance of fusion module design, even as the language features within the model may primarily serve a regularization role.

In conclusion, the enhancements to the edge detector resulting from language feature integration can be attributed to two primary factors: Firstly, it likely contributes regularization as an additional signal to single-modality data regression. Secondly, the most significant improvements stem from the careful design of the fusion module. Without meticulous attention to this aspect, the enhancements remain marginal.

Another notable aspect of integrating language information into edge detection is the model's capacity for zero-shot boundary detection. This involves recognizing text categories not encountered during training, leveraging the adaptable text features from the CLIP text encoder. To achieve this, we modify our model's head network by replacing the convolutional classification head with the LSeg head, performing a matrix multiplication $f_{Fusion} = f_T \cdot f_I$. We select LSeg as our benchmark to showcase the efficacy of our approach. Both models are trained on SBD. We directly incorporate new text labels into the original 20 classes in the SBD dataset. Since this experiment evaluates three new categories [pet, vehicle, rider], the total number of input text categories for the model is 22 (adding "rider" and "vehicle"). The results are depicted in Fig. 7.

In the first example, we replace "horse" with "rider", and in the second, "mbike" becomes "vehicle". These categories and texts are novel to SBD. Our method demonstrates significant superiority, not only in object identification but also in capturing intricate object contours, such as the horse's legs and the motorcycle's storage part—details often overlooked by LSeg. Additionally, our method displays proficiency in complex environments, ensuring dependable identification even in such scenarios.

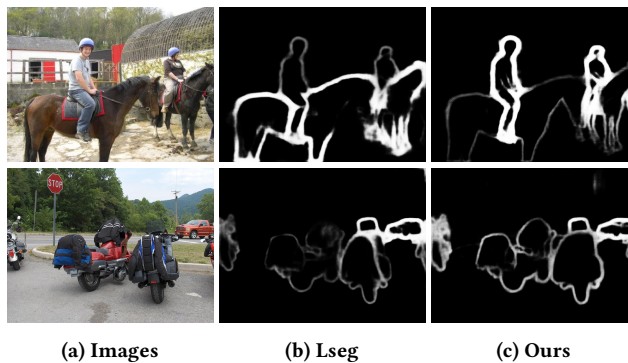

| (a) Images | (b) Lseg | (c) Ours |
|---|---|---|

**Figure 7: Demonstrating zero-shot detection, we replace "horse" with "rider" in one example and "mbike" with "vehicle" in another. These categories and texts are novel to SBD. Our method outperforms LSeg in accurately delineating the boundaries of these new objects.**

### 4.5 Limitation of our work

Although our method exhibits clear advantages over single-modality approaches, there are several drawbacks that require further improvement: 1. The enhancement of high-level image features needs refinement. As observed in the attention maps in Fig. 5, the new attention maps do not fully encompass the objects of interest, with some regions near the object boundaries misdetected. 2. Regarding the feature representation of the final mask encoding, the t-SNE map 6 reveals that our refined mask encoding's clustering results are not as optimal as expected. 3. The dynamic language information generated from LVM and LLM entails large model weights and presents challenges for deployment on edge devices, limiting the practicality of utilizing dynamic language information.

Compared to traditional methods, developing new network architectures for single-modal image data is time-consuming and risks overfitting. Our cross-modal framework, with language assistance for edge detection, is simple, flexible, and robust even in noisy label scenarios. Notably, it doesn't rely on dynamic language information. Our experiments show significant improvement using only static information, distinguishing our framework from SEAL or DDS.

### 5 CONCLUTION

In conclusion, our proposed method addresses the challenges of scene contour detection (SED) by introducing a novel cross-modal approach. Inspired by CLIP-UM and LSeg, our method learns from both images and text to provide comprehensive contour representations. We uniquely tailor our model for SED by incorporating dynamic language information for detailed object descriptions and relationships within each image, and integrating a fusion technique to enhance cross-modality information integration, resulting in refined high-level image features and mask encodings. The experiments illustrate the superiority and robustness of our method compared to previous state-of-the-art (SOTA) methods.

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
