# OpenReview forum: "Advancing Semantic Edge Detection through Cross-Modal Knowledge Learning"
_acmmm.org/ACMMM/2024/Conference — MM2024 Poster_

### Official Review · Reviewer_199Q · 2024-05-23

**Rating:** 4
**Confidence:** 2

**Summary:**

This paper focuses on semantic edge detection (SED) and presents a cross-modal contour detection method for SED. Specifically, they leverage text information to enhance semantic edge detection through a novel cross-modality framework. Extensive experiments significantly improves SED accuracy compared to previous models.

**Strengths:**

1. Clear writing and easy to understand.
2. Experimental results show the best performance.

**Limitations:**

1. The paper introduces that they utilize the LVM to describe the input images, but there are not enough examples to show the response from LVM, which is hard to estimate this method.

2. The method of comparison is not sufficient, and it is too old, and the experimental results may not be very convincing.

3. Have you tried other fusion methods? The current one seems a little direct.

**Suitability:**

2

---

### Official Review · Reviewer_NSaa · 2024-05-23

**Rating:** 3
**Confidence:** 3

**Summary:**

The authors focus on the semantic edge detection task, trying to overcome the challenges of the prevalence of low-quality labels. They introduced a solution to bolster SED through the encoding of both language and image data.

**Strengths:**

1. The method is simple and easy to understand
2. The experiments are well-designed and presented clearly.
3. The topic is valuable and suitable for the MM community

**Limitations:**

1. The motivation of the method is not clear enough, compared to the method using "static elements."  What is the "dynamic content" in the proposed method and why did you choose that as intuition to solve the problem?

2. A question about SBD datasets, is why sometimes only some objects in the front scene are labeled, e.g., in Figure 3, the keyboard is not annotated.

3. It is not clear what the main difference is between "language-driven" edge detection and previous edge detection methods that used text encoders as input. Btw, why it is called language-driven?

**Suitability:**

3

---

### Official Review · Reviewer_sRRV · 2024-06-04

**Rating:** 4
**Confidence:** 1

**Summary:**

The author presents a paper on Semantic Edge Detection (SED), introducing an innovative approach that enhances SED by integrating both language and image data. Unlike previous language-driven techniques that mainly rely on static elements such as dataset labels, their method leverages dynamic language content to describe the objects in each image and their relationships. They propose two fusion points within the encoder-decoder model: one at the high-level feature generation stage to enrich feature representation, and another at the final image feature location to improve mask encoding quality, thereby preventing information loss. Experimental evaluations were conducted on benchmark datasets, including SBD and Cityscape.

**Strengths:**

The paper demonstrates impressive experimental results in comparison to other methods for Semantic Edge Detection (SED). Notably, it introduces a fascinating zero-shot edge detection capability derived from textual information. The paper is well-written, systematically introducing each element and thoroughly explaining every aspect of the proposed method. The novelty of this approach lies in its ability to leverage the power of multimodality to enrich visual features, distinguishing it from existing techniques.

**Limitations:**

In Figure 2, it is recommended to include a "frozen" symbol on the text encoder for clarity. Moreover, both Table 1 and Table 2 would benefit from more descriptive captions. The caption for Figure 5 seems to be inaccurate. Additionally, it's pertinent to mention that the ablation study could also be carried out on the EfficientNet-b7 architecture.
Why is the graph information labeled as dynamic when it's precalculated before training begins? Typically, dynamic content evolves over time. Could you clarify the dynamic nature of this information?

**Suitability:**

3

---

### Official Review · Reviewer_UQas · 2024-06-04

**Rating:** 2
**Confidence:** 4

**Summary:**

The paper highlights the role of inadequate pixel-level labels in semantic edge detection (SED), and the consequence of such unreliable annotations affecting the final result. The paper proposes a cross-modal approach to enhance SED by leveraging both image and dynamic language data (text) that encodes the objects and their relationships in each image. The approach seems to make significant performance improvements on benchmark datasets like SBD and Cityscapes.

**Strengths:**

1. The paper proposes a good approach to include textual information that enhances model performance, grounded in empirical previous works like [17, 20].

2. In addition to using text-encoding of the static labels, the paper also adds encoding of objects and its relations generated by using a combination of LVM and LLM.

**Limitations:**

1. Unclear main figure (Fig. 2):  Fusion modules - the yellow boxes are the two fusion points and should be annotated such to make the figure look more clear. The arrows that show the flow of textual features and image features seems to be hanging in the air. They should join their corresponding text feature (green) and image feature (yellow) box. Also, are high-level features of the encoder same as the Image feature (yellow box)? The way it is shown shows that they are different.

2. Generation of good quality Knowledge Graphs -  This relies heavily on the quality of the base LVM model which generally have much lower accuracy than LLMs. Are the generated objects-relations good-enough? A short user-study could have made this clearer.

3. Equation 2 is buggy: The feature-sizes of fused-textual encoding and image encodings are not compatible since they are outputs of different encoders (text encoder module of CLIP for text, and final layer of the image encoder). For ex: the third line in Eq. 2 adds a FC layer to aligned features of image (fI-aligned). This also changes the dimensions. The operations do not reflect what is shown in Fig. 2.

4. Equation 3 is also buggy: Fig. 2 shows (and the paper also mentions this in the main proposal) that static labels and the dynamic labels are combined and then passed through the textual encoder of CLIP. How, then, can fs and fI be combined and also fD and fI? Since only the combined (static+dynamic) embeddings are available for fusion?

5. The fI-CM cross modal feature is used as input to the decoder and it is also added with the final mask encodings as residual connection - this is unclear in the Fig. 2.

6. #511-513: The choice of using EfficientNet for best performance and VGG16 for ablation is counter-intuitive. Ablation is done to know how and which proposed architectural changes affect the performance of the best model.  Therefore, it follows that the ablation should be done on the best performing model.

7. Does the original data (cityscapes, BSD) have SED labels or are they derived from their corresponding semantic labels?

8. The paper in general needs heavy proof-reading, re-structuring and also clarity on many details as mentioned in above points.


**Formatting Comments**

1. Fig. 2 - the colors used for high-level image features are similar to the output generated by the decoder which could be confusing.

2. #403-404: The paper mentions use of block-5 of encoder, however it is left to the readers to decide whether this is from VGG-16 or EfficientNet - this should be made clear. And it seems that VGG16 is only used for ablations.

3. Tables 3 and 4 appear first in discussion in the paper, and therefore should be numbered in this way. Ablation are additional experiments and their tables come later.

**Suitability:**

2

---

### Meta-Review · Area_Chair_tAkq · 2024-07-04

**Recommendation:** Accept (Poster)
**Confidence:** 3

**Metareview:**

The paper presents an innovative approach to Semantic Edge Detection (SED) by integrating textual data with image data, aiming to address the issue of inadequate pixel-level labels. As reported by the reviewers, the comparison methods was initially considered  outdated, potentially affecting the convincingness of the results. More concerns on the methodological soundness were raised. Nonetheless, the rebuttal successfully addressed the raised concerns. The reviewers acknowledged that most of their questions were satisfactorily answered. The remaining limitations noted by other reviewers, regard particularly clarity and presentation. These issues seem to have contributed to some misunderstandings, which further emphasizes the need for improvements in these areas. However, the proposed approach was deemed promising by the reviewers for the Semantic Edge Detection task. Based on the collective feedback from the reviewers and the authors' rebuttal, I thus recommend Acceptance for this paper. In revising the paper, I strongly encourage the authors to focus on improving writing quality, as suggested in the reviews.